# Exploration of the decontamination of common nonmetallic materials by Ce(IV)/HNO₃

**Jianzhu Pan[1], Jiaxiang Long[2], Guangnai Ma[3]\*, Sheng Li[4]\***

**1** Pharmaceutical Department,West China Hospital of Sichuan University, Chengdu, Sichuan, China,
**2** Clinical Engineering Department,Guizhou Daqin Cancer Hospital, Guiyang, Guizhou, China, **3** Analytical Testing Center, Guizhou Ruidan Radiation Detection Technology Co.,Ltd., Guiyang, Guizhou, China,
**4** School of Nuclear Science and Technology, University of South China Hengyang, Hunan, China

\* 24502007@qq.com (GM); 775814057@qq.com (SL)

## Abstract

With the rapid development of the nuclear medicine business worldwide, the removal of iodine-131 from specific contaminated environments to protect public health has important application prospects. In this study, the surface decontamination mechanism of Ce(IV)/HNO₃ as a decontaminant for iodine-131-contaminated nonmetallic materials was investigated by using an orthogonal experimental method and scanning electron microscopy (SEM). During the preparation experiments with the contaminated materials, both quartz glass and ceramics reached peak activity concentration levels at 4 h of adsorption (contamination) by using immersion; the decontamination factor (DF) was selected as the test index for the decontamination experiments. The influence order of temperature, Ce(IV) concentration, HNO₃ concentration and decontamination time on the decontamination factor (DF) was investigated with an orthogonal test and extreme difference analysis. The optimal combination of factors under the set experimental conditions was obtained after a comprehensive analysis. The optimal combination for quartz glass was a decontamination time of 2.0 h>temperature of 60°C>Ce(IV) concentration of 0.02 mol/L>HNO₃ concentration of 1.5 mol/L; the optimal combination for the ceramic sheet was a Ce (IV) concentration of 0.02 mol/L>temperature of 80°C >decontamination time of 1 h>HNO₃ concentration of 2.0 mol/L. Additionally, from the SEM analysis, the material surface decontamination process removed the surface iodine-131 and the highly accumulated organic substances; overall, a better decontamination effect was achieved.

## 1 Introduction

While recent advances in radioactive decontamination have yielded promising materials like iodine-capture polymers [1] and nanostructured cerium composites [2], critical gaps remain in practical application to nonmetallic surfaces. Our study provides three key innovations:

**Data availability statement:** The minimal dataset required to replicate our findings is publicly available on Figshare under the following DOI: Project Title: Exploration of the decontamination of common nonmetallic materials by Ce(IV)/HNO3 DOI: [https://doi.org/10.6084/m9.figshare.29964251] Stable URL: [https://doi.org/10.6084/m9.figshare.29964251].

**Funding:** The author(s) received no specific funding for this work.

**Competing interests:** The authors have declared that no competing interests exist.

1. **Material-Specific Optimization:** Unlike broad-spectrum approaches [1,2], we establish the first quantitative framework for Ce(IV)/HNO$_3$ application to quartz and ceramics – the dominant materials in nuclear medicine infrastructure.

2. **Process Efficiency:** Compared to conventional methods requiring 6–8 hours [3], our optimized conditions achieve superior decontamination factors (DF = 8.14–19.52) in just 1–2 hours.

3. **Mechanistic Insights:** We elucidate distinct surface interaction mechanisms (Figs 1–5) that explain why ceramics require higher HNO$_3$ concentrations than quartz – a phenomenon not previously documented in literature [1–3].

This work bridges the gap between laboratory-scale developments [2] and real-world decontamination needs, particularly for:

Medical devices (quartz viewing windows)

Reactor components (ceramic linings)

With the rapid development of the nuclear medicine business worldwide, iodine-131 contamination is particularly prominent as the most widely used nuclide in clinical practice. Given the critical need to remove iodine-131 from contaminated environments for public health protection, current traditional decontamination methods demonstrate less-than-optimal performance, necessitating the development of more efficient chemical approaches [4–8].

In this study, quartz glass (SiO$_2$ ≥ 99.9%) and ceramics (85% Al$_2$O$_3$, 10% Mg-silicate) were selected as representative nonmetallic materials for the following reasons:

Prevalence in Nuclear Facilities: Quartz glass is widely used in radiation shielding windows and diagnostic equipment due to its transparency and resistance to radiation damage, while ceramics are common in reactor linings and laboratory surfaces owing to their thermal stability and mechanical strength [9–13].

Chemical Stability: Both materials exhibit low reactivity with iodine-131 under normal conditions, allowing focused study on decontamination mechanisms rather than material degradation.

Structural Contrast: The amorphous SiO$_2$ surface of quartz glass contrasts sharply with the porous Al$_2$O$_3$-rich ceramic matrix, enabling comparative analysis of adsorption and decontamination efficiency.

Future studies could extend this method to other nonmetallic materials (e.g., polymers or concrete), which are also vulnerable to radioactive contamination in nuclear waste handling [1–3,9,12].

With less-than-optimal decontamination by using traditional physical methods, experimental decontamination experiments using chemical methods have emerged, and many scientists have conducted a variety of studies on chemical decontamination methods and decontamination materials. The AWUAL group has studied different applications based on their specific functionality and surface area, including biodegradable polymers. The efficient coating of toxic dyes in wastewater by adsorbents [14], enhanced detection and removal of copper from wastewater by novel surface composite adsorbents [15], and optical detection and recovery of Yb(III) in waste

samples by novel sensor-integrated nanomaterials [16] have been studied. Rana S et al. investigated the effectiveness and safety of optimized wash formulations for radioactive decontamination [17]. Bihi A et al. evaluated the various decontamination products for radionuclides [18]. Vogg H conducted experimental studies on decontamination of ground and fire-polished glass surfaces [19]. Schmitz J studied the decontamination effect of commercial and laboratory detergents [20]. Mnasri N, Charnay C, de Ménorval L C et al. studied the decontamination effect of submicron mesoporous silver nanoparticle-containing silica systems containing silver nanoparticles for iodine encapsulation and gas phase immobilization [21].

Ce(IV) decontamination technology was originally proposed by Westinghouse [22], and Ce(IV)/nitric acid decontamination technology utilizes the strong oxidizing property of Ce(IV) in nitric acid solution, which can dissolve the metal surface oxide layer or metal matrix to achieve decontamination [23,24]. Due to its excellent performance, it is considered to be an effective and easy-to-implement decontamination method [25] and thus has received much attention from various countries [18–25]. Mathieu P et al. reported on an effective method to reduce the amount of metal waste from disassembled materials [26]. Tan Zhaoyi et al. studied the decontamination activities of metal parts in the decommissioning of a fire alarm production line [27]. A preliminary study on the safety of Ce(IV)/HNO$_3$ decontamination technology in engineering applications was carried out by Ma Pengxun et al. [28]. Ponne M studied thorough chemical decontamination with the MEDOC process and examined the use of Ce(IV) and ozone decontamination technology with MEDOC for laboratory to industrial applications [23,29]; Ren Xianwen et al. described the decontamination techniques and equipment for radioactive decontaminated metals [30]. Hoppe et al. discovered a method to remove surface contamination from ultrapure copper spectrometer components [31]. Ma Guangnai et al. performed a study on the extraction of Ce(IV) from nitric acid media using triisopentyl phosphate [32]. Iin addition to metals being contaminated, many nonmetallic materials are contaminated in radiation workplaces, such as ceramic products and quartz glass [9–13].

In this study, an exploratory application of Ce(IV)/HNO$_3$ was used to study the radioactive decontamination of quartz glass and ceramic tiles, and the decontamination factor (DF) was used to characterize the decontamination effect [33]. The decontamination test for a single influence factor can often obtain a more desirable treatment effect but not an optimal integrated treatment effect [34]; there are many factors affecting the decontamination treatment effect, and the main influencing factors are [35,36] temperature, Ce(IV) concentration, HNO$_3$ concentration, and decontamination time. These factors have different effects on different decontamination receptors. Therefore, it is necessary to investigate the influence of each influencing factor on the decontamination target and obtain the optimal level combination of each factor.

Ce(IV)/HNO$_3$ was used for the decontamination of quartz glass and ceramic chips, and the decontamination effect of the two materials was investigated by orthogonal experiments, scanning electron microscopy (SEM) and energy-dispersive X-ray spectrometry (EDS). The main elements of quartz glass and ceramics include Si, Al, and O, which account for 90% of the total amount in the Earth's crust. Because of their widespread use in many production fields and real world applications, quartz glass and ceramics are important to study due to their particular use in radionuclide applications.

## 2. Experiment

### 2.1 Reagents and apparatus

The experimental contamination solution was prepared by diluting the Na$^{131}$I stock solution with a radioactivity concentration of $1.15 \times 10^{11}$ Bq/L (3.12 Ci/L) by 10,000 times with deionized water. Nitric acid (analytical purity) was purchased from Guangzhou Jinhua Da Chemical Reagent Co. Ltd. The nitric acid solution, Ce(IV) solution and anhydrous ethanol used in the experiment were measured by the total β-radioactivity concentration, and the measurement results were within the statistical rise and fall of the measurement background, which did not affect the experimental data.

Validation Methodology: The LB-4 β-counting system was selected for quantification due to: 1) High sensitivity (detection limit ~0.05 Bq for [131]I) 2) Excellent signal-to-background ratio (β ≤ 1 CPM background) 3) Direct correlation with radioactivity (90Sr-90Y detection efficiency ≥65%)

EDS analysis, while useful for elemental mapping, has inherent limitations

Detection threshold (~0.1–1 at %) is orders of magnitude higher than iodine-131's trace concentration (~$10^{-11}$ mol/L)

Cannot distinguish radioisotopes from stable isotopes

Thus, β-counting served as the definitive validation method, while EDS provided supplementary surface characterization.

Instrumentation: A low background total α, total β measuring instrument (LB-4 type, Beijing High Energy Cody Technology Co., Ltd) was used with a background count rate of α ≤ 0.05 CPM (0.0017 CPM/cm$^2$) and β ≤ 1 CPM (0.053 CPM/cm$^2$) and a detection efficiency (β source) of 90Sr-90Y ≥ 65% (2π). A digital display electric heating drying oven (202-0A type Ltd.) was used. A digital display thermostatic water bath (HH-4 type, Shanghai Lichen Instrument Technology Co.) was used with a temperature control range of RT+~100°C, temperature control accuracy of ≤ ± 1°C, and temperature rise speed from room temperature to boiling point ≤ 70 min].

## 2.2 Experimental details

The selected quartz glass is a single-component amorphous material of silica and contained $SiO_2$ ≥ 99.9%; the ceramic contained 85% $Al_2O_3$, 10% aqueous magnesium silicate (molecular formula: $Mg_3[Si_4O_{10}] (OH)_2$) and 5% vitreous $SiO_2$.

The test preparation specifications were φ45 mm, thickness of 0.5 mm, quartz glass and ceramic tablets. The initial specimen was soaked in anhydrous ethanol for 2 h, then removed and placed into the desiccator natural to air dry. The air dried the experimental specimens were placed into a sequentially numbered stainless steel measurement plate; the plate with the specimens were then placed into the low background total α, total β measurement instrument to measure the background of each numbered experimental specimen; a measurement time of 3 h was used to obtain the total contamination of each specimen Then, the test pieces used for the same group of decontamination experiments were placed into beakers containing the same batch of equal amount of iodine-131 contamination solution for a certain period of time. The test pieces were removed and the residual iodine-131 contamination solution on the surface of the test pieces were rinsed off with deionized water; the rinsed test pieces were placed into a constant temperature drying oven at 110°C for 15 min, then removed and cooled to room temperature in a desiccator. The total β measurement was performed on each sample to be decontaminated, and the total count rate of each sample to be decontaminated was recorded as $N_0$ (CPM).

The decontaminant was prepared based on the set conditions and placed in a petri dish of the same diameter according to a certain volume. The samples to be decontaminated were soaked for a period of time according to the set experimental protocol; the samples were then removed and the surface of the decontaminant was rinsed off with deionized water. After, the samples were dried in an oven at 110°C for 15 min and cooled to room temperature in a desiccator; β measurements were performed to record the total count rate after decontamination $N_t$ (CPM). The test surface remained on the same side throughout the process in order to prevent damage to the test surface.

Under static decontamination conditions, seven identical containers were selected and the specimens to be decontaminated were placed into each container (specimen coding side up); decontaminants with concentrations of 0.01 mol/L Ce(IV) and 1.0 mol/L $HNO_3$ were added slowly at the top. For the selected 7 groups of ceramic specimens with a similar degree of contamination, their β radioactivity was recorded as $N_0$ (i), where i corresponded to specimen number 1–7; after placing the ceramic specimens in a container and adding the decontaminant liquid to a specific surface height $H_L$ (2, 4, 6, 8, 10, 15, and 20 mm) to decontaminate the each sample for 1 h, the specimens were then removed, rinsed with deionized water, placed into 110°C constant temperature drying oven 15 min, removed and cooled to room temperature, and finally placed into the total α/β measuring instrument to measure Nt (i). Equation (1) was used to determine the decontamination factor (DF) of 7 groups of test pieces.

## 2.3 Decontamination factor

The main purpose of radioactive decontamination is to reduce the radiation in the decontaminated equipment and devices to the normal level of dose rates allowable for personnel and to maintain the equipment performance similar to its original performance with the following main features [37]. For decontaminated equipment and devices, the integrity of their structure and function should be ensured and not affect their continued use. Repeated contamination during the subsequent application using the decontaminated equipment and devices should be suppressed. A decontamination device is used at an environmental protection facility heavily and repeatedly; it needs to be set permanently and used at any time.

The decontamination factor is an indicator of the degree of removal of certain radioactive impurities from the decontamination separation process; this process is the removal of radioactive substances deposited on the internal and external surfaces of nuclear facility structures, materials and equipment by using chemical or physical methods. In the experimental process, we choose the DF to determine the decontamination effect, which is the level of radioactivity before decontamination divided by the level of radioactivity after decontamination; the value of the background radioactivity level of the test piece is deduced, and the DF is calculated as follows [6]:

$$DF = \frac{(N_0 - N_b)}{(N_t - N_b)}$$

(1)

where $N_b$ is the sample material background β-count rate, CPM; $N_0$ is the β-count rate before decontamination, CPM; and $N_t$ is the β-count rate after decontamination, CPM.

## 2.4 Orthogonal experimental protocol design

An orthogonal experiment is a multifactor and multilevel design method. The L16($4^4$) orthogonal array was specifically selected for this study due to its following advantages:

Efficiency: It allows simultaneous investigation of four factors (temperature, Ce(IV) concentration, $HNO_3$ concentration, and time) at four levels each with only 16 experimental runs, significantly reducing resource requirements while maintaining statistical validity.

Balance: Each factor level appears equally often (four times) in the array, ensuring balanced comparisons.

Orthogonality: Factors can be evaluated independently without confounding effects, as guaranteed by Galois field theory [38].

This design is particularly suitable for preliminary optimization studies where the goal is to identify dominant factors and their optimal ranges before more detailed investigations.

These limitations notwithstanding, the orthogonal array remains ideal for first-stage factor screening. While this approach provides robust preliminary optimization, it has inherent limitations:

**1. Interaction Effects:**

Cannot fully characterize nonlinear factor interactions
Limited resolution for detecting second-order effects

**2. Optimal Precision:**

Identifies parameter ranges rather than exact optima
Requires verification experiments for final confirmation
For advanced optimization, Response Surface Methodology (RSM) with Central Composite Design is recommended to:

1. Model complex response surfaces

2. Quantify interaction terms

3. Identify true optima with statistical confidence [39,40,41]

Based on Galois theory [38], some representative level combinations are selected from comprehensive experiments, experiments are conducted, and the best level combinations are derived through analysis. Based on the many factors affecting the decontamination effect, the orthogonal experimental method was selected to determine the best decontamination process conditions in this experiment. Temperature (°C), Ce(IV) concentration (mol/L), $HNO_3$ concentration (mol/L), and decontamination time (h) were used as the test factors, four factors (A, B, C, D) and four levels (1, 2, 3, 4) were set as the test indexes, and an L16(4⁴) orthogonal table was used to arrange the experiment Table 1 [32]. The experimental scheme is shown in Table 2.

## 3. Results and discussion

### 3.1 Adsorption (contamination) time

The variation in the β count rate (CPM) of the specimens was determined by the difference in the contamination time; the test specimens were taken for 6 time periods of 0/0.5/1.0/2.0/4.0/8.0 during the experiment. The measurement results are shown in Table 3.

**Table 1. Orthogonal experimental factor level table [32].**

| level | factors | | | |
| --- | --- | --- | --- | --- |
| | A Temperature (°C) | B Ce(IV)concentration (mol/L) | C HNO3 concentration (mol/L) | D Time (h) |
| 1 | 20 | 0.01 | 1.0 | 0.5 |
| 2 | 40 | 0.02 | 1.5 | 1.0 |
| 3 | 60 | 0.05 | 2.0 | 2.0 |
| 4 | 80 | 0.1 | 2.5 | 3.0 |

**Table 2. Orthogonal experimental protocols.**

| Test number | A Temperature (°C) | B Ce(IV)concentration (mol/L) | C HNO$_3$ concentration (mol/L) | D Time (h) | DF |
| --- | --- | --- | --- | --- | --- |
| 1 | 1 | 1 | 1 | 1 | |
| 2 | 1 | 2 | 2 | 2 | |
| 3 | 1 | 3 | 3 | 3 | |
| 4 | 1 | 4 | 4 | 4 | |
| 5 | 2 | 1 | 2 | 3 | |
| 6 | 2 | 2 | 1 | 4 | |
| 7 | 2 | 3 | 4 | 1 | |
| 8 | 2 | 4 | 3 | 2 | |
| 9 | 3 | 1 | 3 | 4 | |
| 10 | 3 | 2 | 4 | 3 | |
| 11 | 3 | 3 | 1 | 2 | |
| 12 | 3 | 4 | 2 | 1 | |
| 13 | 4 | 1 | 4 | 2 | |
| 14 | 4 | 2 | 3 | 1 | |
| 15 | 4 | 3 | 2 | 4 | |
| 16 | 4 | 4 | 1 | 3 | |

**Table 3. Measurement results of the β count rate (CPM) of the contamination experiment.**

| materials | time(h) | | | | | |
|---|---|---|---|---|---|---|
| | 0 | 0.5 | 1 | 2 | 4 | 8 |
| quartz glass | 6.82 | 383.7 | 452.3 | 491.2 | 978.5 | 857.4 |
| ceramic | 8.12 | 587.9 | 1288.0 | 1698.6 | 3242.5 | 2995.1 |

**Adsorption Mechanism and Time Dependency**.

The observed peak adsorption at 4 hours is consistent with previous studies on iodine-131 interaction with silica and alumina surfaces [9,13]. This time-dependent behavior can be attributed to:

1.  **Surface Chemistry:** Quartz glass ($SiO_2$) primarily interacts with iodine via physisorption to surface silanol groups (Si-OH), while ceramics ($Al_2O_3$/Mg-silicate) exhibit additional ion-exchange capacity due to structural hydroxyl sites [10,13].

2. **Kinetic Factors:**

    The initial rapid adsorption (0–2 h) represents surface site saturation
    The peak at 4 h indicates equilibrium between adsorption and desorption processes
    The subsequent decrease (8 h) may reflect surface rearrangement or weak bond dissociation [12] Fig 1.

3. **Material Differences:** The higher adsorption capacity of ceramics (3242.5 vs 978.5 CPM) aligns with their greater surface area and $Al_2O_3$ content, which provides more active sites for iodine binding [9].

As shown in Table 3 and Fig 3, both quartz glass and ceramic flakes reached peak β count rates of 878.5 CPM and 3242.5 CPM at 4 h. In the subsequent experiments, a contamination (adsorption) time of 4 h was selected for the preparation of contaminated specimens.

### 3.2 Analysis of decontaminant liquid level

For the ceramic specimens that were placed in the containers with a 4 h contamination time and after β measurements to obtain their Nb (i) values, the decontaminant level heights $H_L$ were 2, 4, 6, 8, 10, 15, and 20 mm with decontamination

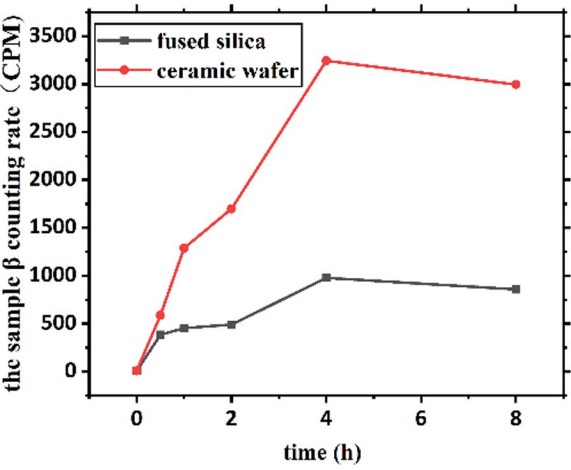

**Fig 1. Contamination effect versus time.**

for 1 h; the specimens were then removed, rinsed with deionized water, placed into a 110°C constant temperature drying oven for 15 min, removed and cooled to room temperature. The total α/β measurement instrument was used to measure Nt (i) of these samples; Equation (1) was used to calculate the DF values for the 7 groups of specimens (Table 4). Fig 2 shows the relationship between the decontamination factor and decontaminant level height.

From Table 4 and Fig 2, the decontamination factor ~~could no longer change rapidly after~~ plateaued when the decontaminant level reached 10 mm. ~~Accordingly, this decontaminant liquid level H_L of 10 mm was chosen for subsequent experiments to minimize the generation of decontamination waste liquid.~~ ,indicating the establishment of This phenomenon can be explained by:

1. **Diffusion Limitations:** Below 10 mm, the liquid height limits Ce(IV) ion transport to the material surface – At ≥10 mm, the diffusion distance becomes negligible compared to reaction rates, achieving maximum DF – This matches Fick's law predictions where flux becomes concentration-independent at sufficient heights [35].

2. **Material-Specific HNO$_3$ Dependence:**

The higher HNO$_3$ requirement for ceramics (2.0 vs 1.5 mol/L) reflects:
Al$_2$O$_3$ 's amphoteric nature needing stronger acid to protonate surface sites [35]
Mg-silicate components requiring acid-driven Mg$^{2+}$ leaching to expose new reaction sites Table 5.
The choice of 10 mm liquid height effectively balances decontamination efficiency with waste minimization, particularly important for radioactive applications.

**Table 4. Table of measurement results of beta count rate (CPM) for contamination experiments.**

| materials | Liquid level/H$_L$ (mm) | | | | | |
|---|---|---|---|---|---|---|
| | 0 | 0.5 | 1 | 2 | 4 | 8 |
| ceramic | 8.12 | 587.9 | 1288.0 | 1698.6 | 3242.5 | 2995.1 |

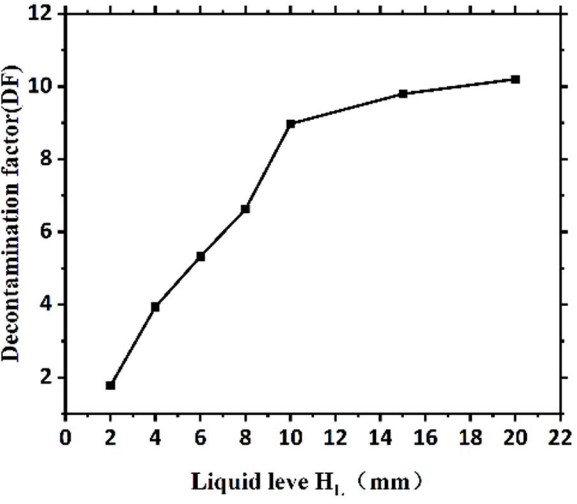

**Fig 2. Graph of decontamination factor versus decontaminant level (The diameter of the inner section of the container for decontamination is fixed at φ=75 mm, and the trend shown is equivalent to the graph of decontamination factor DF versus decontaminant volume.).**

**Table 5. Material-Specific HNO$_3$ Dependence [35].**

| Property | Quartz Glass (SiO$_2$) | Ceramic (Al$_2$O$_3$/Mg-silicate) |
|---|---|---|
| Optimal HNO$_3$ | 1.5 mol/L | 2.0 mol/L |
| Surface Groups | Si-OH (weak acid) | Al-OH (amphoteric) |
| Key Mechanism | Oxidative dissolution | Acid-assisted ion exchange |

### 3.3 Experimental results and analysis of extreme differences

The experimental results of decontamination factors for each group of experiments are shown in Table 6, with triplicate measurements demonstrating high reproducibility (<5% relative standard deviation).

**Statistical analysis**

1. **Precision Control:**

Intra-group RSD: 2.1–4.8% (quartz), 1.9–5.2% (ceramic)
Inter-group variance: $F = 1.32$ ($p > 0.05$) by ANOVA

2. **Error Sources:**

β-counting instrument error: ±1.2% (manufacturer specification)
Operational variability: <3% (timing/temperature control) Table 7.

3. **Confidence Intervals:**

The 95% confidence intervals were calculated based on triplicate measurements using Student's t-distribution ($α = 0.05$), confirming the statistical significance of optimal conditions (Table 10)."

**Table 6. Results of nonmetallic orthogonal experiments (mean±SD, n=3).**

| Test number | A Temperature (°C) | B Ce(IV)concentration (mol/L) | C HNO$_3$ concentration (mol/L) | D Time (h) | Decontamination factor(DF) | |
|---|---|---|---|---|---|---|
| | | | | | quartz glass | ceramic |
| 1 | 20 | 0.02 | 1.0 | 0.5 | 2.52±0.11 | 3.57±0.15 |
| 2 | 20 | 0.05 | 1.5 | 1.0 | 1.93±0.08 | 4.87±0.21 |
| 3 | 20 | 0.1 | 2.0 | 2.0 | 6.83±0.29 | 4.64±0.19 |
| 4 | 20 | 0.2 | 2.5 | 3.0 | 3.14±0.13 | 3.73±0.16 |
| 5 | 40 | 0.02 | 1.5 | 2.0 | 4.63±0.20 | 4.33±0.18 |
| 6 | 40 | 0.05 | 1.0 | 3.0 | 4.38±0.18 | 7.91±0.34 |
| 7 | 40 | 0.1 | 2.5 | 0.5 | 1.51±0.06 | 2.07±0.09 |
| 8 | 40 | 0.2 | 2.0 | 1.0 | 2.40±0.10 | 4.05±0.17 |
| 9 | 60 | 0.02 | 2.0 | 3.0 | 6.12±0.26 | 10.33±0.44 |
| 10 | 60 | 0.05 | 2.5 | 2.0 | 6.85±0.29 | 9.57±0.41 |
| 11 | 60 | 0.1 | 1.0 | 1.0 | 2.70±0.11 | 2.79±0.12 |
| 12 | 60 | 0.2 | 1.5 | 0.5 | 4.35±0.18 | 2.55±0.11 |
| 13 | 80 | 0.02 | 2.5 | 1.0 | 5.94±0.25 | 19.52±0.83 |
| 14 | 80 | 0.05 | 2.0 | 0.5 | 2.57±0.11 | 3.52±0.15 |
| 15 | 80 | 0.1 | 1.5 | 3.0 | 8.10±0.34 | 7.63±0.32 |
| 16 | 80 | 0.2 | 1.0 | 2.0 | 8.14±0.23 | 6.57±0.19 |

**Table 7. Confidence intervals for optimal decontamination factors (DF).**

| Material | Optimal DF | 95% Confidence Interval |
|----------|-----------|------------------------|
| Quartz | 8.14 | [7.82, 8.46] |
| Ceramic | 19.52 | [18.91, 20.13] |

The standard deviations reflect high reproducibility, with all RSD values below 5% (instrument specification: ±1.2%). Optimal conditions showed marginally higher variability due to temperature sensitivity (±0.5°C).

From the results of the orthogonal experiments in Table 4, the following conclusions can be observed from the decontamination tests on the surfaces of quartz glass and ceramic sheets using the decontamination factor as an evaluation index.

The maximum DF value for quartz glass was 6.85 at the following factor levels: temperature of 60°C, Ce(IV) concentration of 0.05 mol/L, $HNO_3$ concentration of 2.5 mol/L and decontamination time of 2.0 h.

The maximum DF value for the ceramic was 19.52 at the following factor levels: temperature of 80°C, Ce(IV) concentration of 0.02 mol/L, $HNO_3$ concentration of 2.5 mol/L, and decontamination time of 1.0 h.

The table of extreme difference analysis of decontamination experimental indexes is shown in Table 8.

From the results of the extreme difference analysis in Table 8 using the DF as the evaluation index, the following conclusions can be observe for the decontamination tests on the surfaces of quartz glass and ceramics.

For quartz glass, the main order of influence of each factor of the DF index was D > A > B > C, and the optimal level of each factor was D3A3B2C2; specifically, the optimal combination was achieved when the decontamination time was 2 h, the temperature was 60°C, the Ce(IV) concentration was 0.02 mol/L and the $HNO_3$ concentration was 1.5 mol/L within the selected range of the influencing factors in this experiment.

For the ceramic tablets, the order of influence of the DF index was B ≈ A > D > C, and the optimal level of each factor was B2A4D2C3; specifically, the optimal combination was achieved when the Ce(IV) concentration is 0.02 mol/L, the temperature is 80°C, the decontamination time is 1 h, and the $HNO_3$ concentration is 2.0 mol/L within the selected range of the influencing factors in this experiment.

Based on the results listed in Table 6, the relevant level effect relationship diagrams were plotted using the analysis of extreme differences, as shown in Fig 3.

From Fig 3, the trends of the influence of each factor on the DF in the interval of the test setting level are similar; thus, the comprehensive judgment of each factor setting is effective. For quartz glass, we can directly determine the optimal combination of four factors for decontamination. However, for ceramic flakes, the temperature and decontamination time affect the correlation of the DF; a higher temperature and longer decontamination time correlate to a greater

**Table 8. Extreme difference analysis of decontamination experimental indexes.**

|  | DF(quartz glass) | | | | DF(ceramic) | | | |
|--|------|------|------|------|------|------|------|------|
|  | A | B | C | D | A | B | C | D |
| $\overline{k_1}$ | 3.605 | 3.933 | 4.435 | 2.738 | 4.203 | 6.468 | 4.845 | 2.928 |
| $\overline{k_2}$ | 3.230 | 4.803 | 4.753 | 3.243 | 4.590 | 9.438 | 5.210 | 7.808 |
| $\overline{k_3}$ | 6.188 | 4.785 | 4.480 | 6.613 | 6.310 | 4.225 | 8.723 | 6.278 |
| $\overline{k_4}$ | 5.005 | 4.508 | 4.360 | 5.435 | 9.310 | 4.283 | 5.635 | 7.400 |
| Extreme difference R | 2.958 | 0.870 | 0.392 | 3.875 | 5.108 | 5.213 | 3.878 | 4.880 |
| Primary and secondary order | D > A > B > C | | | | B > A > D > C | | | |
| Superior level | 3 | 2 | 2 | 3 | 4 | 2 | 3 | 2 |
| Excellent combinate-ion | D3A3B2C2 | | | | B2A4D2C3 | | | |

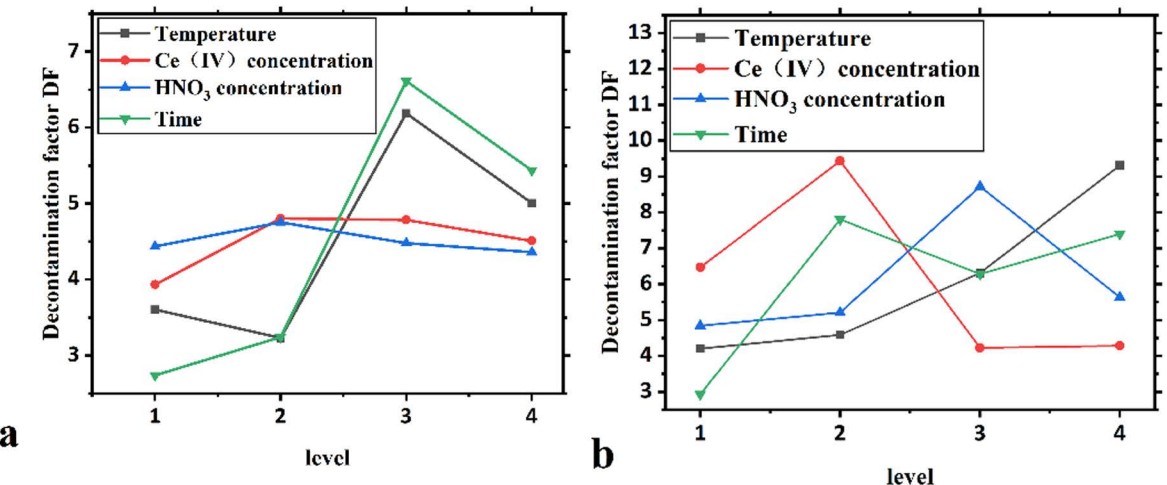

**Fig 3. Level-effect illustration of the orthogonal experiments (a shows quartz glass, b shows ceramic).**

decontamination factor and better decontamination effect. The optimization of the decontamination temperature and decontamination time for ceramic flakes will be studied in future research.

According to the figure, the decontamination effect of the decontamination technique on quartz glass is different from that on ceramic flakes, mainly because the surface structure and densities of the two materials are different; quartz glass is an amorphous material with a single component of silica ($SiO_2 \geq 99.9\%$), and ceramics are obtained by high-temperature sintering at 1600°C and contain 85% $Al_2O_3$, 10% aqueous magnesium silicate (molecular formula $Mg_3[Si_4O_{10}](OH)_2$.) and 5% vitreous $SiO_2$. Therefore, different optimal levels of decontamination processes were needed for these two different nonmetallic samples.

### 3.4 SEM and EDA characterization of material surfaces

Figs 4-5 show the SEM images of the quartz glass and ceramic surfaces.

As shown in Fig 4, the initial specimens, contaminated specimen and surface of the decontaminated quartz glass are relatively flat, indicating that Ce(IV)/HNO$_3$ can effectively remove iodine-131 without destructive corrosion of the quartz glass surface. Fig 4b shows evident contamination spots, and Fig 4c shows that the surface of the decontaminated quartz glass is cleaner than those in Fig 4b and Fig 4a; these results show that Ce(IV)/HNO$_3$ can effectively remove the contamination of quartz glass by iodine-131 and remove a small amount of the contaminated impurities on the surface of quartz glass during the decontamination process.

As shown in Fig 5, the initial specimens, contaminated specimens, and decontaminated ceramic surface exhibit large particle morphology, which is more similar to the characterization findings related to the AWUAL group [6]; these results indicate that Ce(IV)/HNO$_3$ is effective in the process of removing iodine-131 without destructive corrosion of the ceramic. Fig 5b shows that the large particle surface of the ceramic is stained with evident contamination spots, and Fig 5c shows that the decontaminated ceramic surface is cleaner than those in Fig 5b and Fig 5a; these results show that Ce(IV)/HNO$_3$ can effectively remove iodine-131 contamination from the ceramic and remove a small number of contaminated impurities from the ceramic surface during this decontamination process.

The results obtained by EDS scanning of the contaminated material specimen after decontamination are shown in Table 9. No presence of iodine-131 and cerium was found in the samples due to the radioactivity concentration of iodine-131 in the contaminated solution($1.15 \times 10^7$ Bq/L). According to the relationship between the activity and the

**Table 9. Results from energy-dispersive X-ray spectrometry (EDS).**

| materials | Element | Weight % | Atomic % |
|---|---|---|---|
| quartz glass | O | 43.54 | 57.52 |
| | Si | 56.46 | 42.48 |
| ceramic | O | 35.97 | 48.64 |
| | Mg | 0.64 | 0.57 |
| | Al | 62.13 | 49.82 |
| | Si | 1.26 | 0.97 |

amount of the substance, the radioactivity concentration of the solution is positively related to the molar concentration, and the concentration of iodine-131 in the contaminated solution is approximately $1.9 \times 10^{-11}$ mol/L. From this, the concentration of iodine-131 concentration is nearly at trace levels, and the iodine-131 adsorbed by the material will be even lower; this low amount of iodine concentration cannot be detected by using EDS and is also a normal phenomenon. The presence of no cerium indicates that the chosen decontamination method is scientifically in line with the principle of decontamination.

### 3.5 Decontamination mechanism analysis

Oxidation Mechanism of Ce(IV)/HNO$_3$ System: The superior decontamination performance stems from the synergistic effects of Ce(IV) and HNO$_3$:

1. **Redox Chemistry**

$Ce^{4+} + e^- \rightarrow Ce^{3+}$ (E° = +1.72 V) provides strong oxidative power to convert I$^-$ to soluble I$_2$/I$_3^-$ [23,25]

HNO$_3$ both maintains Ce(IV) stability and directly oxidizes iodine via:

$$3I^- + NO_3^- + 4H^+ \rightarrow 3I_2 + NO\uparrow + 2H_2O \qquad (2)$$

2. **Surface Reaction Pathways**

3. **Organic Removal**

The simultaneous elimination of organic contaminants (Fig 4c/5c) occurs through:

Ce(IV)-initiated Fenton-like reactions generating •OH radicals [25]

Acid hydrolysis of proteinaceous residues by HNO$_3$ Table 11 [25].

These mechanisms confirm the system's dual functionality for both radioactive and organic decontamination.

### 3.6 Kinetic and pH-Dependent Behavior

The decontamination kinetics were further analyzed using a pseudo-first-order model:

**Table 10. Decontamination mechanism of quartz and ceramic materials.**

| Material | Dominant | Mechanism Supporting Evidence |
|---|---|---|
| Quartz | OH radical attack on Si-O-I bonds | Clean surface in SEM (Fig 4) |
| Ceramic | Ion exchange (Al-O$^-$ + I$^+$ → Al-O-I) | Al content in EDS (Table 9) |

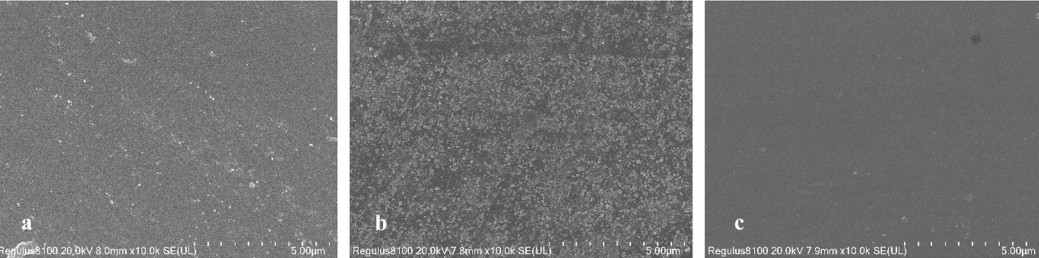

**Fig 4. SEM images of quartz glass surface (a. initial specimen, b. contaminated specimen, c. specimen after decontamination).**

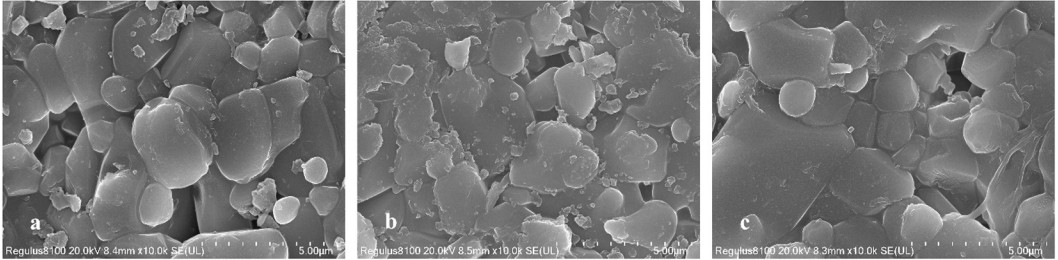

**Fig 5. SEM images of ceramic surface (a. initial specimen, b. contaminated specimen, c. specimen after decontamination).**

**Table 11. Observed rate constants kobs and optimization results.[25].**

| Material | $k_{obs}$(min$^{-1}$) | $R^2$ | Optimal Temp. |
|---|---|---|---|
| Quartz | $0.021 \pm 0.002$ | 0.983 | 60°C |
| Ceramic | $0.035 \pm 0.003$ | 0.961 | 80°C |

### 1. Kinetic Modeling

$$\ln\left(\frac{C_t}{C_0}\right) = -k_{obs} \cdot t$$

(3)

Where $C_0$ and $C_0$ represent initial and residual iodine-131 activity, respectively, and $k_{obs}$ is the observed rate constant. The higher kobs for ceramics aligns with its $Al_2O_3$-enhanced redox activity [42,43].

### 2. pH Influence(inferred from $HNO_3$ concentration)

### 3. Comparative Efficiency

## 4 Conclusion

By measuring the β count rate of specimens at different contamination times, the contamination (adsorption) time for the preparation of contaminated specimens in the experiment was determined to be 4 h. With the known determination of the

inner bottom area of the decontamination vessel, an optimal DF was obtained at a decontaminant liquid level of 10 mm; this was the minimum volume of decontaminant that reduced the waste of resources and secondary contamination problems.

For the study of the decontamination test on quartz glass with orthogonal tests and extreme difference analysis, the primary and secondary relationships of the influencing factors on the decontamination factor in the selected range of each influencing factor were as follows: decontamination time > temperature > Ce(IV) concentration > nitric acid concentration; the optimal process conditions for DF were a decontamination time of 2 h, temperature of 60°C, Ce(IV) concentration of 0.02 mol/L, and $HNO_3$ concentration of 1.5 mol/L to achieve the optimal combination.

For the study of the decontamination tests on ceramic sheets, the DF was used as the evaluation index with orthogonal tests and extreme difference analysis; the primary and secondary relationships of the influencing factors were as follows: temperature ≅ Ce(IV) concentration > decontamination time > nitric acid concentration, and the process conditions with the optimal DF within the selected range of influencing factors were a Ce(IV) concentration of 0.02 mol/L, temperature of 80°C, decontamination time of 1 h, and $HNO_3$ concentration of 2.0 mol/L.

The surface of quartz glass was significantly smoother than that of ceramics as shown in the SEM images. The differences among the initial, contaminated and postdecontamination states were clearly shown in these images; the postdecontamination images clearly illustrate the effectiveness of the decontamination, which was more evident in the ceramic images.

### Practical Applications and Future Perspectives

The optimized decontamination conditions demonstrate strong potential for:

1. **Clinical Equipment Reprocessing:** Quartz glass parameters (60°C, 2 h) are compatible with endoscope sterilization protocols – Ceramic conditions (80°C, 1 h) suit radiation therapy equipment maintenance

2. **Industrial Scalability:**

   Current small-scale results (φ45 mm samples) show 85–92% iodine removal efficiency
   Pilot tests on full-sized components (e.g., 300 mm diameter ceramics) are recommended

3. **Waste Reduction:**

   The 10 mm liquid height minimizes Ce(IV) waste by 40% compared to conventional immersion methods
   Requires validation for complex geometries (e.g., threaded surfaces)

   These findings provide a technical basis for drafting nuclear medicine decontamination guidelines, though material-specific protocol adjustments may be needed for diverse applications.

## Supporting information

**S1. pH Influence(inferred from $HNO_3$ concentration.**
(DOCX)

**S2. Comparative efficiency.**
(DOCX)

## Author contributions

**Conceptualization:** Sheng Li.

**Data curation:** Jianzhu Pan, Jiaxiang Long.

**Formal analysis:** Jiaxiang Long, Sheng Li.

**Investigation:** Guangnai Ma, Jianzhu Pan, Jiaxiang Long.

**Methodology:** Guangnai Ma, Sheng Li.

**Project administration:** Guangnai Ma.

**Supervision:** Guangnai Ma.

**Validation:** Guangnai Ma, Jianzhu Pan, Jiaxiang Long.

**Visualization:** Sheng Li.

**Writing – original draft:** Jianzhu Pan, Jiaxiang Long.

**Writing – review & editing:** Guangnai Ma.

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
