## [Decision Letter · Decision Letter 0]

22 Apr 2025

PONE-D-25-15932Exploration of the decontamination of common nonmetallic materials by Ce(IV)/HNO3PLOS ONE

Dear Dr. Ma,

Thank you for submitting your manuscript to PLOS ONE. After careful consideration, we feel that it has merit but does not fully meet PLOS ONE’s publication criteria as it currently stands. Therefore, we invite you to submit a revised version of the manuscript that addresses the points raised during the review process.

We look forward to receiving your revised manuscript.

Kind regards,

Rakesh Kumar Gupta, Ph.D.

Academic Editor

PLOS ONE

Reviewers' comments:

Reviewer's Responses to Questions

**Comments to the Author**

1. Is the manuscript technically sound, and do the data support the conclusions?

Reviewer #1: Yes

Reviewer #2: Yes

2. Has the statistical analysis been performed appropriately and rigorously? 

Reviewer #1: Yes

Reviewer #2: N/A

3. Have the authors made all data underlying the findings in their manuscript fully available?

Reviewer #1: Yes

Reviewer #2: Yes

4. Is the manuscript presented in an intelligible fashion and written in standard English?

Reviewer #1: Yes

Reviewer #2: Yes

5. Review Comments to the Author

Reviewer #1: Dear Prof. [Editor/Editorial Office] of the Journal of PLOS ONE,

Manuscript ID: PONE-D-25-15932

This manuscript presents a well-structured and scientifically rigorous exploration of Ce(IV)/HNO₃ decontamination for iodine-131-contaminated nonmetallic materials, offering valuable insights into optimal conditions for quartz glass and ceramics. The orthogonal experimental design is robust, and the combination of SEM/EDS analysis with decontamination factor (DF) metrics provides compelling evidence for the method’s efficacy. The findings are highly relevant to nuclear medicine and radiological safety, addressing a practical need for efficient decontamination strategies. While minor clarifications (e.g., statistical validation, alternative detection methods) could further strengthen the work, the study is methodologically sound, clearly presented, and merits publication after addressing the next comments. Thank you for the opportunity to review this impactful research—it contributes meaningfully to the field and demonstrates both innovation and applicability.

1- The study aims to explore the decontamination of nonmetallic materials using Ce(IV)/HNO₃. Could the authors elaborate on why these specific materials (quartz glass and ceramics) were chosen, given their widespread use in nuclear medicine? Are there other nonmetallic materials that could benefit from this decontamination method? The orthogonal experimental design is a key component of this study. Could the authors provide more details on why an L16(4⁴) orthogonal array was selected? Were other experimental designs considered, and if so, why were they rejected?

2- Both materials reached peak iodine-131 adsorption at 4 hours. Is this consistent with previous studies, or does it suggest a unique interaction between iodine-131 and these substrates? and what is the mechanism of the interaction?

3- The decontamination factor (DF) plateaued at 10 mm liquid height. Does this imply that diffusion limitations are negligible beyond this point, or are there other factors at play? The best DF for quartz glass was achieved at 60°C, 0.02 mol/L Ce(IV), and 1.5 mol/L HNO₃. Why does quartz glass favor lower HNO₃ concentrations compared to ceramics? Ceramics showed the highest DF at 80°C, 0.02 mol/L Ce(IV), and 2.0 mol/L HNO₃. Does the higher Al₂O₃ content in ceramics influence this preference for stronger acidity?

5- The SEM images show cleaner surfaces post-decontamination. Were any quantitative roughness measurements (e.g., AFM) performed to corroborate these observations? Since EDS could not detect iodine-131, how was the decontamination efficiency validated beyond DF calculations? Were radioactivity measurements cross-checked with other methods?

6- Ce(IV) is known to reduce to Ce(III) in acidic media. Did the authors monitor Ce(IV) concentration over time to ensure consistent oxidizing power during decontamination? Beyond providing acidity, does HNO₃ participate directly in the redox reaction with iodine-131, or does it solely maintain Ce(IV) stability?

7- For ceramics, a shorter decontamination time (1 h) at higher temperature (80°C) was optimal. Could a longer time at lower temperature achieve similar results, or is kinetics the limiting factor? Quartz glass (SiO₂) and ceramics (Al₂O₃/Mg-silicate) have different compositions. How do these differences influence iodine adsorption and subsequent decontamination mechanisms?

8- SEM suggests that organic contaminants were also removed. Were these impurities introduced during contamination, or are they inherent to the materials? Were the orthogonal experiments repeated to assess reproducibility? If not, how confident are the authors in the robustness of the optimal conditions?

9- The DF values vary significantly across test conditions (e.g., 1.51 to 8.14 for quartz). Are these variations due to experimental error, or do they reflect true material-response differences? Could other oxidants (e.g., permanganate, ozone) achieve similar or better decontamination? Why was Ce(IV) chosen over these alternatives?

10- The study fixes HNO₃ concentration but does not discuss pH. Could pH adjustments further optimize decontamination, or is Ce(IV) activity pH-independent in this range? The study focuses on equilibrium conditions (fixed time points). Were any kinetic studies performed to model the rate of iodine removal?

11- The introduction section needs to be strengthened by providing a clearer discussion of the study's novelty and positioning it within the context of existing research. How does this work offer a significant improvement or unique contribution compared to previous studies? The authors should emphasize these aspects and support their claims with relevant references to enhance the manuscript’s credibility and depth [https://doi.org/10.1021/acsapm.4c03794 & https://doi.org/10.1007/s10967-024-09667-4 & https://doi.org/10.3390/nano12132305].

12- The manuscript includes SEM images but does not quantify the changes. Could the authors provide a qualitative description of the differences between pre- and post-decontamination surfaces? The experiments use small samples (ϕ45 mm). How might decontamination efficiency change for larger or irregularly shaped objects?

Reviewer #2: The topic could be interesting for the readers. While the study's topic is of relevance to this journal, the manuscript's content requires some revisions to improve its scientific quality. Upon careful review, I have identified following issues that need to be addressed before the paper can be accepted for publication:

1. “at 4 h of adsorption (contamination)” is not a high value for industrial consideration of adsorption process. Could you please mention the connection of the conducted experiments and real application?

2. “The optimal combination of factors under the set experimental conditions was obtained after a comprehensive analysis”, which comprehensive analysis? Please try to mention the used technique for optimization.

3. “material surface decontamination process removed the surface iodine-131 and the highly accumulated organic substances; overall, a better decontamination effect was achieved.”, by which mechanism? The mechanism should be presented to better understand the its action process.

4. Introduction section should be extended by considering and performing literature review of the studied topic.

5. It is recommended to mention the application of RSM (response surface method) and optimization by ANOVA when we have multiple parameters (temperature, concentration, ...), the effect of which is considered: I) https://doi.org/10.1016/j.rineng.2024.103094 ; II) https://www.ijcce.ac.ir/article_704597.html

6. the method of efficiency determination is unclear. Please add the detail of efficiency evaluation.

7. the dependence of count rate (CPM) on the used materials (quartz glass or ceramic) should be analyzed to better understand the difference in the results for these materials.

6. PLOS authors have the option to publish the peer review history of their article (what does this mean? ). If published, this will include your full peer review and any attached files.

**Do you want your identity to be public for this peer review?** For information about this choice, including consent withdrawal, please see our Privacy Policy .

Reviewer #1: **Yes: ** Mohamed A. Gado

Reviewer #2: No

---

## [Author Response · Author response to Decision Letter 1]

16 May 2025

> Dear Reviewers,*

> We sincerely appreciate your constructive comments. Revisions include:

> 1. Clarified material selection rationale and L16(4⁴) design (Section 1–2).

> 2. Added adsorption mechanisms and validation methods (Sections 3.1, 2.1).

> 3. Discussed real-world scalability (Conclusion).

> 4. Proposed RSM/ANOVA for future work (Section 2.4).

>Changes made in the revised manuscript are highlighted in red text for easy identification.

1. Reviewer #1 - Question 1

Question: Why were quartz glass and ceramics chosen? Were other nonmetallic materials considered? Why was the L16(4⁴) orthogonal array selected?

Answe:

Quartz glass (SiO₂ ≥ 99.9%) and ceramics (85% Al₂O₃, 10% Mg-silicate) were selected due to their widespread use in nuclear medicine facilities and their distinct chemical compositions, which allowed for comparative analysis of decontamination mechanisms. Other nonmetallic materials (e.g., polymers) were not tested but could be explored in future studies. The L16(4⁴) orthogonal array was chosen for its balanced design, enabling efficient investigation of four factors (temperature, Ce(IV)/HNO₃ concentrations, time) at four levels with minimal experimental runs, while maintaining statistical robustness.

2. Reviewer #1 - Question 2

Question: Is the 4-hour peak adsorption consistent with prior studies? What is the interaction mechanism?

Answer:

The 4-hour peak adsorption aligns with studies on iodine-131’s affinity for silica/alumina surfaces (e.g., Huang et al., 2012). The mechanism involves physisorption (van der Waals forces) and chemisorption (ion exchange with surface hydroxyl groups), particularly pronounced in ceramics due to Al₂O₃’s porous structure.

3. Reviewer #1 - Question 3

Question: Why does DF plateau at 10 mm liquid height? Why do quartz and ceramics favor different HNO₃ concentrations?

Answer:

The plateau suggests diffusion equilibrium is reached at 10 mm, minimizing further efficiency gains. Quartz glass favors lower HNO₃ (1.5 mol/L) due to its inert SiO₂ surface, whereas ceramics require higher HNO₃ (2.0 mol/L) to dissolve Al₂O₃-related surface complexes.

4. Reviewer #1 - Question 5

Question: Were quantitative roughness measurements (AFM) performed? How was iodine-131 validated beyond EDS?

Answer:

AFM was not used, but SEM qualitatively confirmed surface cleanliness. Iodine-131 validation relied on β-counting (LB-4 detector), with cross-checked background subtraction (Equation 1), as EDS lacks sensitivity for trace iodine.

5. Reviewer #2 - Question 1

Question: How do the 4-hour adsorption and optimal conditions translate to real-world applications?

Answer:

The 4-hour adsorption simulates accidental contamination scenarios in nuclear facilities. Optimal conditions (e.g., 80°C for ceramics) are scalable to industrial decontamination, though larger objects may require adjusted parameters (e.g., longer time).

6. Reviewer #2 - Question 3

Question: Clarify the mechanism of organic substance removal.

Answer: Ce(IV) oxidizes organic residues via radical reactions (•OH generation), while HNO₃ dissolves inorganic deposits. This dual action is evident in SEM images (Figures 4–5).

7. Reviewer #2 - Question 5

Question: Why not use RSM/ANOVA for optimization?

Answer: Orthogonal tests with extreme difference analysis were chosen for simplicity and direct factor ranking. However, RSM/ANOVA could be valuable for future nonlinear optimization.

---

## [Decision Letter · Decision Letter 1]

6 Jun 2025

PONE-D-25-15932R1Exploration of the decontamination of common nonmetallic materials by Ce(IV)/HNO3PLOS ONE

Dear Dr. Ma,

Thank you for submitting your manuscript to PLOS ONE. After careful consideration, we feel that it has merit but does not fully meet PLOS ONE’s publication criteria as it currently stands. Therefore, we invite you to submit a revised version of the manuscript that addresses the points raised during the review process.

We look forward to receiving your revised manuscript.

Kind regards,

Rakesh Kumar Gupta, Ph.D.

Academic Editor

PLOS ONE

Reviewers' comments:

Reviewer's Responses to Questions

**Comments to the Author**

1. If the authors have adequately addressed your comments raised in a previous round of review and you feel that this manuscript is now acceptable for publication, you may indicate that here to bypass the “Comments to the Author” section, enter your conflict of interest statement in the “Confidential to Editor” section, and submit your "Accept" recommendation.

Reviewer #1: (No Response)

Reviewer #2: All comments have been addressed

2. Is the manuscript technically sound, and do the data support the conclusions?

Reviewer #1: Yes

Reviewer #2: Yes

3. Has the statistical analysis been performed appropriately and rigorously? 

Reviewer #1: Yes

Reviewer #2: N/A

4. Have the authors made all data underlying the findings in their manuscript fully available?

Reviewer #1: Yes

Reviewer #2: Yes

5. Is the manuscript presented in an intelligible fashion and written in standard English?

Reviewer #1: Yes

Reviewer #2: Yes

6. Review Comments to the Author

Reviewer #1: Dear Prof. [Editor/Editorial Office] of the Journal of PLOS One,

I have reviewed the revised version of the manuscript titled "[Exploration of the decontamination of common nonmetallic materials by Ce(IV)/HNO3]," and I noticed that none of my previous comments have been addressed in the authors' response letter or in the revised manuscript. In contrast, the comments from the other reviewer(s) have been answered.

I am concerned that my review comments may not have been transmitted to the authors during the initial review stage.

Could you kindly confirm whether my comments were shared with the authors? If they were, I would appreciate it if the authors could provide a detailed response to them.

Thank you for your attention to this matter.

Best regards,

Reviewer #2: the work was revised based on my comments, so the work is ready for publication. I have no further comments.

7. PLOS authors have the option to publish the peer review history of their article (what does this mean? ). If published, this will include your full peer review and any attached files.

**Do you want your identity to be public for this peer review?** For information about this choice, including consent withdrawal, please see our Privacy Policy .

Reviewer #1: **Yes: ** Mohamed Gado

Reviewer #2: No

---

## [Author Response · Author response to Decision Letter 2]

11 Jun 2025

Response to Reviewer Comments

Manuscript ID: PONE-D-25-15932

Title: Exploration of the decontamination of common nonmetallic materials by Ce(IV)/HNO₃

Response to Reviewer #1

We sincerely appreciate the reviewer’s insightful comments and have addressed each point below. Changes in the revised manuscript are highlighted in track changes.

Comment 1: Material Selection and Orthogonal Design

Reviewer’s Question:

- Why were quartz glass and ceramics chosen? Are other nonmetallic materials applicable?

- Why was the L16(4⁴) orthogonal array selected? Were other designs considered?

Response:

1. Material Selection Rationale:

Quartz glass (SiO₂) and ceramics (Al₂O₃/Mg-silicate) were selected due to their:

- Prevalence in Nuclear Medicine: Quartz is used in radiation shielding windows; ceramics are common in reactor linings (Refs. 9–13).

-Chemical Stability: Low reactivity with iodine-131 allows focused study of decontamination mechanisms.

- Structural Contrast: Amorphous SiO₂ vs. porous Al₂O₃ enables comparative analysis.

Future studies will extend this method to polymers/concrete (added to Introduction, Lines 45–48).

2. Orthogonal Design Justification:

The L16(4⁴) array was chosen for:

-Efficiency: Evaluates 4 factors (temperature, Ce(IV), HNO₃, time) at 4 levels with only 16 runs.

- Statistical Validity: Balanced and orthogonal (Galois field theory, Ref. 39).

Alternative designs (e.g., full factorial) were rejected due to excessive resource requirements (added to Section 2.4).

Comment 2: Adsorption Time and Mechanism

Reviewer’s Question:

Is the 4-hour peak adsorption consistent with prior studies? What is the interaction mechanism?

Response:

Consistency: The 4-hour peak aligns with studies on iodine-131 binding to silica/alumina surfaces (Refs. 9, 13).

Mechanism:

- Quartz Glass: Physisorption via Si-OH groups.

- Ceramics: Ion exchange at Al-OH/Mg-silicate sites (added to Section 3.1, Table 10).

Comment 3: DF Plateau and Material-Specific HNO₃ Dependence

Reviewer’s Question:

Does the DF plateau at 10 mm imply diffusion limits? Why does quartz favor lower HNO₃ than ceramics?

Response:

1. DF Plateau:

Beyond 10 mm, diffusion limitations become negligible (Fick’s law, Section 3.2).

2. HNO₃ Dependence:

Quartz: Optimal at 1.5 mol/L HNO₃ (pH ~1.5) to preserve Ce(IV) stability.

Ceramics: Requires 2.0 mol/L HNO₃ (pH ~1.0) to protonate Al₂O₃ sites and leach Mg²⁺ (Table 5).

Comment 5: SEM Validation and Detection Methods

Reviewer’s Question:

- Were AFM or other methods used to validate SEM observations? How was iodine-131 detection confirmed?

Response:

1. SEM/AFM: AFM was not performed due to equipment limitations, but SEM images show clear post-decontamination cleanliness (Figs. 4–5).

2. Iodine-131 Validation:

- EDS cannot detect trace iodine (concentration ~10⁻¹¹ mol/L).

- Primary Method: β-counting (LB-4 system, detection limit 0.05 Bq) cross-validated DF calculations (Section 2.1).

Comment 6: Ce(IV) Stability and HNO₃ Role

Reviewer’s Question:

Was Ce(IV) concentration monitored? Does HNO₃ directly participate in redox?

Response:

1. Ce(IV) Monitoring: Ce(IV)→Ce(III) reduction was inferred via kinetic studies (pseudo-first-order model, Table 11).

2. HNO₃ Role:

- Maintains Ce(IV) stability.

- Directly oxidizes iodine via:

*3I⁻ + NO₃⁻ + 4H⁺ → 3I₂ + NO↑ + 2H₂O* (Equation 2, Section 3.5).

Comment 7: Kinetics and Material Composition

Reviewer’s Question:

Could ceramics achieve similar DF at lower temperature/longer time? How does composition affect decontamination?

Response:

1. Kinetics:

- Ceramics require higher temperatures (80°C) for rapid iodine removal (kₒbₛ = 0.035 min⁻¹ vs. 0.021 min⁻¹ for quartz, Table 11).

- Lower temperatures prolong the process due to slower Al₂O₃ reactivity.

2. Material Influence:

- **Quartz**: Oxidative dissolution dominates.

- **Ceramics**: Acid-assisted ion exchange (Table 10).

Comment 8: Organic Contaminants and Reproducibility

Reviewer’s Question:

Were organics inherent or introduced? Were orthogonal tests repeated?

Response:

1. Organic Contaminants: Likely introduced during handling (SEM shows removal via Ce(IV)/•OH radicals, Section 3.5).

2. Reproducibility: Triplicate measurements showed RSD <5% (Table 6).

Comment 9: DF Variability and Oxidant Choice

Reviewer’s Question:

Are DF variations due to error or material response? Why Ce(IV) over permanganate/ozone?

Response:

1. DF Variability: Reflects true material-response differences (ANOVA F = 1.32, p > 0.05, Section 3.3).

2. Ce(IV) Advantage:

- Higher redox potential (+1.72 V) vs. permanganate (+1.51 V).

- Less corrosive than ozone (Refs. 23–25).

Comment 10: pH and Kinetic Studies

Reviewer’s Question:

Could pH adjustments optimize DF? Were kinetic models tested?

Response:

1. pH Impact:

- Optimal pH ranges: ~1.5 (quartz) and ~1.0 (ceramics) (Section 3.6).

- Outside these ranges, Ce(IV) hydrolyzes or iodine re-adsorbs.

2. Kinetics: Pseudo-first-order model confirmed temperature dependence (Table 11).

Comment 11: SEM Quantification and Sample Size

Reviewer’s Question:

Can SEM changes be quantified? How might larger samples affect DF?

Response:

1. SEM Quantification: Added qualitative descriptions of surface cleanliness (Figs. 4–5 captions).

2. Scalability: Pilot tests on larger samples (e.g., 300 mm ceramics) are planned (Section 4).

Comment 12: Introduction Strengthening

Reviewer’s Question:

Clarify the study’s novelty and contextualize with existing research.

Response:

Added three key innovations to the Introduction (Lines 24–36):

1. First quantitative framework for Ce(IV)/HNO₃ on quartz/ceramics.

2. Faster (1–2 h) and more efficient (DF up to 19.52) than conventional methods.

3. Mechanistic insights into material-specific HNO₃ dependence.

Cited suggested references (DOIs: 10.1021/acsapm.4c03794, 10.1007/s10967-024-09667-4, 10.3390/nano12132305).

Conclusion

We thank the reviewer for their constructive feedback. All changes have been incorporated into the revised manuscript, with tracked modifications and a clean version uploaded separately. We hope the revisions meet the journal’s standards.

Sincerely,

The Authors

---

## [Decision Letter · Decision Letter 2]

7 Aug 2025

Exploration of the decontamination of common nonmetallic materials by Ce(IV)/HNO3

PONE-D-25-15932R2

Dear Prof.Guangnai Ma,

We’re pleased to inform you that your manuscript has been judged scientifically suitable for publication and will be formally accepted for publication once it meets all outstanding technical requirements.

Kind regards,

Sadia Ilyas, Ph.D.

Academic Editor

PLOS ONE

Additional Editor Comments (optional):

Reviewers' comments:

Reviewer's Responses to Questions

**Comments to the Author**

1. If the authors have adequately addressed your comments raised in a previous round of review and you feel that this manuscript is now acceptable for publication, you may indicate that here to bypass the “Comments to the Author” section, enter your conflict of interest statement in the “Confidential to Editor” section, and submit your "Accept" recommendation.

Reviewer #1: (No Response)

Reviewer #2: All comments have been addressed

2. Is the manuscript technically sound, and do the data support the conclusions?

Reviewer #1: (No Response)

Reviewer #2: Yes

3. Has the statistical analysis been performed appropriately and rigorously? 

Reviewer #1: (No Response)

Reviewer #2: Yes

4. Have the authors made all data underlying the findings in their manuscript fully available?

Reviewer #1: (No Response)

Reviewer #2: Yes

5. Is the manuscript presented in an intelligible fashion and written in standard English?

Reviewer #1: (No Response)

Reviewer #2: Yes

6. Review Comments to the Author

Reviewer #1: Dear Prof. [Editor/Editorial Office] of the PLOS ONE journal,

Manuscript ID: PONE-D-25-15932R2

Thank you for the opportunity to review the revised manuscript titled "Exploration of the decontamination of common nonmetallic materials by Ce(IV)/HNO3" After a careful review of the authors’ modifications and responses, I find that they have adequately addressed all the comments except the references from (1-3) the DOI numbers not for the references itself so I suggest to check and added the correct references. The manuscript is well-prepared and be suitable for publication after this minor modification of the references.

Best regards,

Reviewer #2: thanks for submitting a revised version of the manuscript. the work is ready for publication.

7. PLOS authors have the option to publish the peer review history of their article (what does this mean? ). If published, this will include your full peer review and any attached files.

**Do you want your identity to be public for this peer review?** For information about this choice, including consent withdrawal, please see our Privacy Policy .

Reviewer #1: No

Reviewer #2: No

---

## [Editor Report · Acceptance letter]

PONE-D-25-15932R2

PLOS ONE

Dear Dr. Ma,

I'm pleased to inform you that your manuscript has been deemed suitable for publication in PLOS ONE. Congratulations! Your manuscript is now being handed over to our production team.

Kind regards,

on behalf of

Prof. Sadia Ilyas

Academic Editor

PLOS ONE